# IMITATION LEARNING BY REINFORCEMENT LEARNING

**Kamil Ciosek**
Spotify
kamilc@spotify.com

## ABSTRACT

Imitation learning algorithms learn a policy from demonstrations of expert behavior. We show that, for deterministic experts, imitation learning can be done by reduction to reinforcement learning with a stationary reward. Our theoretical analysis both certifies the recovery of expert reward and bounds the total variation distance between the expert and the imitation learner, showing a link to adversarial imitation learning. We conduct experiments which confirm that our reduction works well in practice for continuous control tasks.

## 1 INTRODUCTION

Typically, reinforcement learning (RL) assumes access to a pre-specified reward and then learns a policy maximizing the expected average of this reward along a trajectory. However, specifying rewards is difficult for many practical tasks (Atkeson & Schaal, 1997; Zhang et al., 2018; Ibarz et al., 2018). In such cases, it is convenient to instead perform Imitation Learning (IL), learning a policy from expert demonstrations.

There are two major categories of Imitation Learning algorithms: behavioral cloning and inverse reinforcement learning. Behavioral cloning learns the policy by supervised learning on expert data, but is not robust to training errors, failing in settings where expert data is limited (Ross & Bagnell, 2010). Inverse reinforcement learning (IRL) achieves improved performance on limited data by constructing reward signals and calling an RL oracle to maximize these rewards (Ng et al., 2000).

The most versatile IRL method is adversarial IL (Ho & Ermon, 2016; Li et al., 2017; Ghasemipour et al., 2020), which minimizes a divergence between the distribution of data produced by the agent and provided by the expert. Adversarial IL learns a representation and a policy simultaneously by using a non-stationary reward obtained from a discriminator network. However, training adversarial IL combines two components which are hard to stabilize: a discriminator network, akin to the one used in GANs, as well as a policy, typically learned with an RL algorithm with actor and critic networks. This complexity makes the training process very brittle.

There is a clear need for imitation learning algorithms that are simpler and easier to deploy. To address this need, Wang et al. (2019) proposed to reduce imitation learning to a single instance of reinforcement learning problem, where reward is defined to be one for state-action pairs from the expert trajectory and zero for other state-action pairs. A closely related, but not identical, algorithm has been proposed by Reddy et al. (2020) (we describe the differences in Section 5). However, while empirical performance of these approaches has been good, they enjoy no performance guarantees at all, even in the asymptotic setting where expert data is infinite.

**Contributions**   We fill in this missing justification for this algorithm, providing the needed theoretical analysis. Specifically, in Sections 3 and 4, we show a total variation bound between the expert policy and the imitation policy, providing a high-probability performance guarantee for a finite dataset of expert data and linking the reduction to adversarial imitation learning algorithms. For stochastic experts, we describe how the reduction fails, completing the analysis. Moreover, in Section 6, we empirically evaluate the performance of the reduction as the amount of available expert data varies.

## 2 PRELIMINARIES

**Markov Decision Process**   An average-reward Markov Decision Process (Puterman, 2014; Feinberg & Shwartz, 2012) is a tuple $(S, A, T, R, s_1)$, where $S$ is a state space, $A$ is the action space, $T : S \times A \to P(S)$ is the transition model, $R : S \times A \to [0, 1]$ is a bounded reward function and $s_1$ is the initial state. Here, we write $P(X)$ to denote probability distributions over a set $X$. A stationary policy $\pi : S \to P(A)$ maps environment states to distributions over actions. A policy $\pi$ induces a Markov chain $P_\pi$ over the states. In the theoretical part of the paper, we treat MDPs with finite state and action spaces. Given the starting state $s_1$ of the MDP, and a policy $\pi$, the limiting distribution over states is defined as

$$\rho_S^\pi = \mathbb{1}(s_1)^\top \lim_{N \to \infty} \frac{1}{N} \sum_{i=0}^N P_\pi^i, \tag{1}$$

where $\mathbb{1}(s_1)$ denotes the indicator vector. We adopt the convention that the subscript $S$ indicates a distribution over states and no subscript indicates a distribution over state-action pairs. We denote $\rho^\pi(s, a) = \rho_S^\pi(s)\pi(a|s)$. While the limit in equation 1 is guaranteed to exist for all finite MDPs (Puterman, 2014), without requiring ergodicity, in this paper we consider policies that induce ergodic chains. Correspondingly, our notation for $\rho_S^\pi$ and $\rho^\pi$ does not include the dependence on the initial state. The expected per-step reward of $\pi$ is defined as

$$V^\pi = \lim_{N \to \infty} \mathbb{E}_\pi [J_N/N \mid s_1] = \mathbb{E}_{\rho^\pi}[R(s, a)], \tag{2}$$

where $J_N = \sum_{t=1}^N R(s_t, a_t)$ is the total return. In this paper we consider undiscounted MDPs because they are enough to model the essential properties of the imitation learning problem while being conceptually simpler than discounted MDPs. However, we believe that similar results could be obtained for discounted MDPs.

**Expert Dataset**   Assume that the expert follows an ergodic policy $\pi_E$. Denote the corresponding distribution over expert state-action pairs as $\rho^E(s, a) = \rho^{\pi_E}(s, a)$, which does not depend on the initial state by ergodicity. Consider a finite trajectory $s_1, a_1, s_2, a_2, \ldots, s_N, a_N$. Denote by $\hat{\rho}(s, a) = \frac{1}{N} \sum_{t=1}^N \mathbb{1}\{(s, a) = (s_t, a_t)\}$ the histogram (empirical distribution) of data seen along this trajectory and denote by $\hat{\rho}_S(s) = \frac{1}{N} \sum_{t=1}^N \mathbb{1}\{s = s_t\}$ the corresponding histogram over the states. Denote by $D$ the support of $\hat{\rho}$, i.e. the set of state-action pairs visited by the expert.

**Total Variation**   Consider probability distributions defined on a discrete set $X$. The total variation distance between distributions is defined as

$$\|\rho_1 - \rho_2\|_{\text{TV}} = \sup_{M \subseteq X} |\rho_1(M) - \rho_2(M)|. \tag{3}$$

In our application, the set $X$ is either the set of MDP states $S$ or the set of state state-action pairs $S \times A$. Below, we restate in our notation standard results about the total variation distance. See Appendix D for proofs.

**Lemma 1.** *If $\mathbb{E}_{\rho_1}[v] - \mathbb{E}_{\rho_2}[v] \leq \epsilon$ for any vector $v \in [0, 1]^{|X|}$, then $\|\rho_1 - \rho_2\|_{\text{TV}} \leq \epsilon$.*

**Lemma 2.** *If $\|\rho_1 - \rho_2\|_{\text{TV}} \leq \epsilon$, then $\mathbb{E}_{\rho_1}[v] - \mathbb{E}_{\rho_2}[v] \leq 2\epsilon$ for any vector $v \in [0, 1]^{|X|}$.*

**Imitation Learning**   Learning to imitate means obtaining a policy that mimics expert behavior. This can be formalized in two ways. First, we can seek an imitation policy $\pi_I$ which obtains a expected per-step reward that is $\epsilon$-close to the expert on any reward signal bounded in the interval $[0, 1]$. Denote the limiting distribution of state-action tuples generated by the imitation learner with $\rho^I(s, a)$. Formally, we want to satisfy

$$\forall R : S \times A \to [0, 1], \quad \mathbb{E}_{\rho^E}[R] - \mathbb{E}_{\rho^I}[R] \leq \epsilon, \tag{4}$$

where $\epsilon$ is a small constant. Second, we can seek to ensure that the distributions of state-action pairs generated by the expert and the imitation learner are similar (Ho & Ermon, 2016; Finn et al., 2016). In particular, if we measure the divergence between distributions with total variation, we want to ensure

$$\|\rho^E - \rho^I\|_{\text{TV}} \leq \epsilon. \tag{5}$$

---

**Algorithm 1** Imitation Learning by Reinforcement Learning (ILR)

---

**Require:** expert dataset $D$, ENVIRONMENT (without access to extrinsic reward)
  $R_{\text{int}}(s, a) \leftarrow \mathbb{1}\{(s, a) \in D\}$
  $\pi_I \leftarrow \text{RL-SOLVER}(\text{ENVIRONMENT}, R_{\text{int}})$

---

Lemmas 1 and 2 show that the equations 4 and 5 are closely related. In other words, attaining expected per-step reward close to the expert's is the same (up to a multiplicative constant) as closeness in total variation between the state-action distribution of the expert and the imitation learner. We provide further background on adversarial imitation learning and other divergences in Section 5.

**Markov Chains**  When coupled with the dynamics of the MDP, a policy induces a Markov chain, In this paper, we focus our attention on chains which are irreducible and aperiodic. Crucially, such chains approach the stationary distribution at an exponential rate. In the following Lemma, we formalize this insight. The proof is given in Appendix D.

**Lemma 3.** *Denote with $P_E$ the Markov chain induced by an irreducible and aperiodic expert policy. The mixing time of the chain $\tau_{\text{mix}}$, defined as the smallest $t$ so that $\max_{s'} \|\mathbb{1}(s')^\top P_E^t - \rho_S^E\|_{\text{TV}} \le \frac{1}{4}$ is bounded. Moreover, for any probability distribution $p_1$, we have $\sum_{t=0}^{\infty} \|p_1^\top P_E^t - \rho_S^E\|_{\text{TV}} \le 2\tau_{\text{mix}}$.*

## 3 IMITATION LEARNING BY REINFORCEMENT LEARNING

**Algorithm**  To perform imitation learning, we first obtain an expert dataset $D$ and construct an intrinsic reward

$$R_{\text{int}}(s, a) = \mathbb{1}\{(s, a) \in D\}. \tag{6}$$

We can then use any RL algorithm to solve for this reward. We note that for finite MDPs equation 6 exactly matches the algorithm proposed by Wang et al. (2019) as a heuristic. The reward defined in equation 6 is intrinsic, meaning we construct it artificially in order to elicit a certain behavior from the agent. This is in contrast to an extrinsic reward, which is what the agent gathers in the real world and what we want to recover. We will learn in Proposition 1 that the two are in fact closely related and solving for the intrinsic reward guarantees recovery of the extrinsic reward.

**Guarantee on Imitation Policy**  The main focus of the paper lies on showing theoretical properties of the imitation policy obtained when solving for the reward signal defined in equation 6. Our formal guarantees hold under three assumptions.

**Assumption 1.** *The expert policy is deterministic.*

**Assumption 2.** *The expert policy induces an irreducible and aperiodic Markov chain.*

**Assumption 3.** *The imitation learner policy induces an irreducible and aperiodic Markov chain.*

Assumption 1 is critical for our proof to go through and cannot be relaxed. It is also essential to rationalize the approach of Wang et al. (2019). In fact, no reduction involving a single call to an RL solver with stationary reward can exist for stochastic experts since for any MDP there is always an optimal policy which is deterministic. Assumptions 2 and 3 could in principle be relaxed, to allow for periodic chains, but it would complicate the reasoning, which we wanted to avoid. We note that we only interact with the expert policy via a finite dataset of $N$ samples.

Our main contribution is Proposition 1, in which we quantify the performance of our imitation learner. We state it below and prove it in Section 4.

**Proposition 1.** *Consider an imitation learner trained on a dataset of size $N$, which attains the limiting distribution of state-action pairs $\rho^I$. Under Assumptions 1, 2 and 3, given that we have $N \ge \max\{800|S|, \ 450 \log\left(\frac{2}{\delta}\right)\} \tau_{\text{mix}}^3 \eta^{-2}$ expert demonstrations, with probability at least $1 - \delta$, the imitation learner attains total variation distance from the expert of at most*

$$\|\rho^E - \rho^I\|_{\text{TV}} \le \eta. \tag{7}$$

*The constant $\tau_{\text{mix}}$ is the mixing time of the expert policy and has been defined in Section 2. Moreover, with the same probability, for any bounded extrinsic reward function $R$, the imitation learner achieves*

*expected per-step extrinsic reward of at least*

$$\mathbb{E}_{\rho^I}[R] \geq \mathbb{E}_{\rho^E}[R] - \eta. \tag{8}$$

Proposition 1 shows that the policy learned by imitation learner satisfies two important desiderata: we can guarantee both the closeness of the generated state-action distribution to the expert distribution as well as recovery of expert reward. Moreover, the total variation bound in equation 7 links the obtained policy to adversarial imitation learning algorithms. We describe this link in more detail in Section 5.

## 4 PROOF

**Structure of Results**   We begin by showing a result about mixing in Markov chains in Section 4.1. We then give our proof, which has two main parts. In the first part, in Section 4.2, we show that it is possible to achieve a high expected per-step intrinsic reward defined in equation 6. In the second part, in Section 4.3, we show that, achieving a large expected per-step intrinsic reward guarantees a large expected per-step extrinsic reward for any extrinsic reward bounded in $[0, 1]$. In Section 4.4, we combine these results and prove Proposition 1.

### 4.1 MIXING IN MARKOV CHAINS

We now show a lemma that quantifies how fast the histogram approaches the stationary distribution. The proof of the lemma relies on standard results about the Total Variation distance between probability distributions as well as generic results about mixing in MDPs, developed by Paulin (2015).

**Lemma 4.** *The total variation distance between the expert distribution and the expert histogram $\hat{\rho}$ based on $N$ samples can be bounded as*

$$\|\rho^E - \hat{\rho}\|_{\mathrm{TV}} \leq \epsilon + \sqrt{\frac{8|S|\tau_{\mathrm{mix}}}{N}} \tag{9}$$

*with probability at least $1 - 2\exp\left\{-\frac{\epsilon^2 N}{4.5\tau_{mix}}\right\}$ (where the probability space is defined by the process generating the expert histogram).*

*Proof.* We first prove the statement for histograms over the states. Recall the notation $\hat{\rho}_S = \frac{1}{N}\sum_{t=1}^N \mathbb{1}\{s_t = s\}$. Recall that we denote the stationary distribution (over states) of the Markov chain induced by the expert with $\rho_S^E$.

First, instantiating Proposition 3.16 of Paulin (2015), we obtain

$$\mathbb{E}_{\hat{\rho}_S}[\|\rho_S^E - \hat{\rho}_S\|_{\mathrm{TV}}] \leq \sum_s \min\left(\sqrt{\frac{4\rho_S^E(s)}{N\gamma_{\mathrm{ps}}}}, \rho_S^E(s)\right),$$

where $\gamma_{\mathrm{ps}}$ is pseudo spectral gap of the chain. Re-write the right-hand side as

$$\sum_s \min\left(\sqrt{\frac{4\rho_S^E(s)}{N\gamma_{\mathrm{ps}}}}, \rho_S^E(s)\right) \leq \sum_s \sqrt{\frac{4\rho_S^E(s)}{N\gamma_{\mathrm{ps}}}} \leq \sqrt{\frac{4|S|}{N\gamma_{\mathrm{ps}}}},$$

where the last inequality follows from the Cauchy-Schwartz inequality. Using equation 3.9 of Paulin (2015), we have $\gamma_{\mathrm{ps}} \geq \frac{1}{2\tau_{\mathrm{mix}}}$. Taken together, this gives

$$\mathbb{E}_{\hat{\rho}_S}[\|\rho_S^E - \hat{\rho}_S\|_{\mathrm{TV}}] \leq \sqrt{\frac{8|S|\tau_{\mathrm{mix}}}{N}}. \tag{10}$$

Instantiating Proposition 2.18 of Paulin (2015), we obtain

$$P(|\|\rho_S^E - \hat{\rho}_S\|_{\mathrm{TV}} - \mathbb{E}_{\hat{\rho}_S}[\|\rho_S^E - \hat{\rho}_S\|_{\mathrm{TV}}]| \geq \epsilon) \leq 2\exp\left\{-\frac{\epsilon^2 N}{4.5\tau_{\mathrm{mix}}}\right\}. \tag{11}$$

Combining equations 10 and 11, we obtain

$$P\left(|\|\rho_S^E - \hat{\rho}_S\|_{\mathrm{TV}} \geq \epsilon + \sqrt{\frac{8|S|\tau_{\mathrm{mix}}}{N}}\right) \leq 2\exp\left\{-\frac{\epsilon^2 N}{4.5\tau_{\mathrm{mix}}}\right\}.$$

Since the expert policy is deterministic, the total variation distance between the distributions of state-action tuples and states is the same, i.e. $\|\rho_S^E - \hat{\rho}_S\|_{\mathrm{TV}} = \|\rho^E - \hat{\rho}\|_{\mathrm{TV}}$. $\qquad\square$

## 4.2 High Expected Intrinsic Per-Step Reward is Achievable

We now want to show that it is possible for the imitation learner to attain a large intrinsic reward. Informally, the proof asks what intrinsic reward the expert would have achieved. We then conclude that a learner specifically optimizing the intrinsic reward will obtain at least as much.

**Lemma 5.** *Generating an expert histogram with $N$ points, with probability at least $1 - 2\exp\left\{-\frac{\epsilon^2 N}{4.5\tau_{mix}}\right\}$, we obtain a dataset such that a policy $\pi_I$ maximizing the intrinsic reward $R_{\text{int}}(s,a) = \mathbb{1}((s,a) \in D)$ satisfies $\mathbb{E}_{\rho^I}[R_{\text{int}}] \geq 1 - \epsilon - \sqrt{\frac{8|S|\tau_{\text{mix}}}{N}}$, where we used the shorthand notation $\rho^I = \rho^{\pi_I}$ to denote the state-action distribution of $\pi_I$.*

*Proof.* We invoke Lemma 4 obtaining $\|\rho^E - \hat{\rho}\|_{\text{TV}} \leq \epsilon + \sqrt{\frac{8|S|\tau_{\text{mix}}}{N}}$ with probability as in the statement of the lemma. First, we prove

$$\sum_{s,a} \rho^E(s,a)\mathbb{1}\left\{\hat{\rho}(s,a) = 0\right\} \leq \epsilon + \sqrt{\frac{8|S|\tau_{\text{mix}}}{N}}, \tag{12}$$

by setting $\rho_1 = \rho^E$, $\rho_2 = \hat{\rho}$ and $M = \{(s,a) : \hat{\rho}(s,a) = 0\}$ in equation 3.

Combining

$$1 = \sum_{s,a} \rho^E(s,a) = \sum_{s,a} \rho^E(s,a)\mathbb{1}\left\{\hat{\rho}(s,a) = 0\right\} + \sum_{s,a} \rho^E(s,a)\mathbb{1}\left\{\hat{\rho}(s,a) > 0\right\} \tag{13}$$

and equation 12, we obtain

$$\sum_{s,a} \rho^E(s,a)\mathbb{1}\left\{\hat{\rho}(s,a) > 0\right\} \geq 1 - \epsilon - \sqrt{\frac{8|S|\tau_{\text{mix}}}{N}}, \tag{14}$$

which means that the expert policy achieves expected per-step intrinsic reward of at least $1 - \epsilon - \sqrt{\frac{8|S|\tau_{\text{mix}}}{N}}$. This lower bounds the expected per-step reward obtained by the optimal policy. □

## 4.3 Maximizing Intrinsic Reward Leads to High Per-Step Extrinsic Reward

We now aim to prove that the intrinsic and extrinsic rewards are connected, i.e. maximizing the intrinsic reward leads to a large expected per-step extrinsic reward. The proofs in this section are based on the insight that, by construction of intrinsic reward in equation 6, attaining intrinsic reward of one in a given state implies agreement with the expert. In the following Lemma, we quantify the outcome of achieving such agreement for $\ell$ consecutive steps.

**Lemma 6.** *Assume an agent traverses a sequence of $\ell$ state-action pairs, in a way consistent with the expert policy. Denote the expert's expected extrinsic per-step reward with $\mathbb{E}_{\rho^E}[R]$. Denote by $\tilde{\rho}_t^{p_1}(s,a) = (p_1^\top P_E^t)(s)\pi_E(a|s)$ the expected state-action occupancy at time $t$, starting in state distribution $p_1$. The per-step extrinsic reward $\tilde{V}_{p_1}^\ell$ of the agent in expectation over realizations of the sequence satisfies*

$$\tilde{V}_{p_1}^\ell = \sum_{s,a}\left(\left(\sum_{t=0}^{\ell-1}\frac{1}{\ell}\tilde{\rho}_t^{p_1}(s,a)\right)R(s,a)\right) \geq \mathbb{E}_{\rho^E}[R] - \frac{4\tau_{\text{mix}}}{\ell}. \tag{15}$$

*Proof.* Invoking Lemma 2, we have that $\mathbb{E}_{\rho^E}[R] - \mathbb{E}_{\tilde{\rho}_t^{p_1}}[R] \leq 2\|\tilde{\rho}_t^{p_1} - \rho^E\|_{\text{TV}}$ for any timestep $t$. In other words, the expected per-step reward obtained in step $t$ of the sequence is at least $\mathbb{E}_{\rho^E}[R] - 2\|\tilde{\rho}_t^{p_1} - \rho^E\|_{\text{TV}}$. The per-step reward in the sequence is at least

$$\tilde{V}_{p_1}^\ell \geq \frac{1}{\ell}\sum_{i=0}^{\ell-1}(\mathbb{E}_{\rho^E}[R] - 2\|\tilde{\rho}_i^{p_1} - \rho^E\|_{\text{TV}}) = \mathbb{E}_{\rho^E}[R] - \frac{2}{\ell}\sum_{i=0}^{\ell-1}\|\tilde{\rho}_i^{p_1} - \rho^E\|_{\text{TV}}$$
$$\geq \mathbb{E}_{\rho^E}[R] - \frac{2}{\ell}\sum_{i=0}^{\infty}\|\tilde{\rho}_i^{p_1} - \rho^E\|_{\text{TV}}$$

Invoking Lemma 3, we have that $\sum_{t=0}^{\infty}\|p_1^\top P_E^t - \rho_S^E\|_{\text{TV}} \leq 2\tau_{\text{mix}}$. Since the expert policy is deterministic, distances between distributions of states and state-action pairs are the same. We thus get $\tilde{V}_{p_1}^\ell \geq \mathbb{E}_{\rho^E}[R] - \frac{4\tau_{\text{mix}}}{\ell}$.

□

In Lemma 6, we have shown that, on average, agreeing with the expert for a number of steps guarantees a certain level of extrinsic reward. We will now use this result to guarantee extrinsic reward obtained over a long trajectory.

**Lemma 7.** *For any extrinsic reward signal $R$ bounded in $[0, 1]$, an imitation learner which attains expected per-step intrinsic reward of $\mathbb{E}_{\rho^I}[R_{\text{int}}] = 1 - \kappa$ also attains extrinsic per-step reward of at least*

$$\mathbb{E}_{\rho^I}[R] \geq (1 - \kappa)(\mathbb{E}_{\rho^E}[R]) - 4\tau_{\text{mix}}\kappa,$$

*with probability one, where $\mathbb{E}_{\rho^E}[R]$ is the expected per-step extrinsic reward of the expert.*

We provide the proof in Appendix D.

## 4.4 BRINGING THE PIECES TOGETHER

We now show how Lemma 5 and Lemma 7 can be combined to obtain Proposition 1 (stated in Section 3).

*Proof of Proposition 1.* We use the assumption that $N \geq \max\left\{800|S|,\ 450 \log\left(\frac{2}{\delta}\right)\right\} \tau_{\text{mix}}^3 \eta^{-2}$. Since $\tau_{\text{mix}}$ is either zero or $\tau_{\text{mix}} \geq 1$, this implies

$$N \geq \max\left\{32|S|\tau_{\text{mix}}\left(1 + 4\tau_{\text{mix}}\right)^2 \eta^{-2},\ \log\left(\frac{2}{\delta}\right) 18\tau_{\text{mix}}\left(1 + 4\tau_{\text{mix}}\right)^2 \eta^{-2}\right\}. \tag{16}$$

We will instantiate Lemma 5 with

$$\epsilon = \frac{\eta}{2 + 8\tau_{\text{mix}}}. \tag{17}$$

Equations 16 and 17 imply that $N \geq \log\left(\frac{2}{\delta}\right) 18\tau_{\text{mix}}\left(1 + 4\tau_{\text{mix}}\right)^2 \eta^{-2} = \log\left(\frac{2}{\delta}\right) 4.5\tau_{\text{mix}}\epsilon^{-2}$. We can rewrite this as $\log\left(\frac{2}{\delta}\right) \leq \frac{N\epsilon^2}{4.5\tau_{\text{mix}}}$, which implies $-\frac{\epsilon^2 N}{4.5\tau_{\text{mix}}} \leq \log\frac{\delta}{2}$. This implies that

$$\delta \geq 2 \exp\left\{-\frac{\epsilon^2 N}{4.5\tau_{\text{mix}}}\right\}. \tag{18}$$

Moreover, using equation 16 again, we have that $N \geq 32|S|\tau_{\text{mix}}\left(1 + 4\tau_{\text{mix}}\right)^2 \eta^{-2}$. This is equivalent to $\frac{8|S|\tau_{\text{mix}}}{N} \leq \frac{1}{4}\frac{\eta^2}{(1 + 4\tau_{\text{mix}})^2}$, or $\sqrt{\frac{8|S|\tau_{\text{mix}}}{N}} \leq \frac{1}{2}\frac{\eta}{1 + 4\tau_{\text{mix}}}$. Using equation 17, we obtain $\epsilon + \sqrt{\frac{8|S|\tau_{\text{mix}}}{N}} \leq \frac{\eta}{1 + 4\tau_{\text{mix}}}$, equivalent to $\left(\epsilon + \sqrt{\frac{8|S|\tau_{\text{mix}}}{N}}\right)(1 + 4\tau_{\text{mix}}) \leq \eta$. Rewriting this gives

$$\kappa\left(1 + 4\tau_{\text{mix}}\right) \leq \eta, \tag{19}$$

where we use the notation $\kappa = \epsilon + \sqrt{\frac{8|S|\tau_{\text{mix}}}{N}}$.

We first show equation 8, which states that we are able to recover expected per-step expert reward. Invoking Lemma 5 and using equation 18, we have that it is possible for the imitation learner to achieve per-step expected intrinsic reward of at least $1 - \kappa$ with probability $1 - \delta$. Invoking Lemma 7, this implies achieving extrinsic reward of at least

$$\mathbb{E}_{\rho^I}[R] \geq \mathbb{E}_{\rho^E}[R](1 - \kappa) - 4\tau_{\text{mix}}\kappa.$$

This implies

$$\mathbb{E}_{\rho^E}[R] - \mathbb{E}_{\rho^I}[R] \leq \mathbb{E}_{\rho^E}[R] - \mathbb{E}_{\rho^E}[R](1 - \kappa) + 4\tau_{\text{mix}}\kappa$$
$$= \kappa\mathbb{E}_{\rho^E}[R] + 4\tau_{\text{mix}}\kappa \leq \kappa + 4\tau_{\text{mix}}\kappa \leq \eta,$$

where the last inequality follows from equation 19. Since this statement holds for any extrinsic reward signal $R$, we obtain the total variation bound by invoking Lemma 1, getting $\|\rho^E - \rho^I\|_{\text{TV}} \leq \eta$. $\quad\square$

## 5 RELATED WORK

**Behavioral Cloning**   The simplest way of performing imitation learning is to learn a policy by fitting a Maximum Likelihood model on the expert dataset. While such 'behavioral cloning' is easy to implement, it does not take into account the sequential nature of the Markov Decision Process, leading to catastrophic compounding of errors (Ross & Bagnell, 2010). In practice, the performance of policies obtained with behavioral cloning is highly dependent on the amount of available data. In contrast, our reduction avoids the problem faced by behavioral cloning and provably works with limited expert data.

**Apprenticeship Learning**   Apprenticeship Learning (Abbeel & Ng, 2004) assumes the existence of a pre-learned linear representation for rewards and then proceeds in multiple iterations. In each iteration, the algorithm first computes an intrinsic reward signal and then calls an RL solver to obtain a policy. Our algorithm also uses an RL solver, but unlike Apprenticeship Learning, we only call it once.

**Adversarial IL**   Modern adversarial IL algorithms (Ho & Ermon, 2016; Finn et al., 2016; Li et al., 2017; Sun et al., 2019) remove the requirement to provide a representation by learning it online. They work by minimizing a divergence between the expert state-action distribution and the one generated by the RL agent. For example, the GAIL algorithm (Ho & Ermon, 2016) minimizes the Jensen-Shannon divergence. It can be related to the total variation objective of equation 5 in two ways. First, the TV and JS divergences obey $2\sqrt{D_{\text{JS}}(\rho^E, \rho^I)} \geq \|\rho^E - \rho^I\|_{\text{TV}}$, so that $D_{\text{JS}}(\rho^E, \rho^I) \leq \epsilon'$ implies $\|\rho^E - \rho^I\|_{\text{TV}} \leq 2\sqrt{\epsilon'}$. This means that GAIL indirectly minimizes the total variation distance. Second, we have $\|\rho^E - \rho^I\|_{\text{TV}} \geq D_{\text{JS}}(\rho^E, \rho^I)$. Since our algorithm minimizes Total Variation for sufficiently large sizes of the expert dataset, this means that it also minimizes the GAIL objective. Both of these properties imply that our reduction converges to a fixpoint with similar properties as GAIL's. This has huge practical significance because our algorithm does not need to train a discriminator.

Another adversarial algorithm, InfoGAIL (Li et al., 2017) minimizes the 1-Wasserstein divergence, which obeys $\mathbb{E}_{\rho^E}[R_L] - \mathbb{E}_{\rho^I}[R_L] \leq D_{\text{W}^1}(\rho^E, \rho^I)$ for any reward function $R_L$ Lipschitz in the $L_1$ norm. This is similar to the property defined by equation 4, implied by our Proposition 1, except we guarantee closeness of expected per-step reward for any bounded reward function as opposed to Lipschitz-continuous functions.

**Random Expert Distillation**   Wang et al. (2019) propose an algorithm which, for finite MDPs, is the same as ours. However, they do not justify the properties of the imitation policy formally. Our work can be thought of as complementary. While Wang et al. (2019) conducted an empirical evaluation using sophisticated support estimators, we fill in the missing theory. Moreover, while our empirical evaluation is smaller in scope, we attempt to be more complete, demonstrating convergence in cases where only partial trajectories are given.

**The SQIL heuristic**   SQIL (Reddy et al., 2020) is close to our work in that it proposes a similar algorithm. At any given time, SQIL performs off-policy reinforcement learning on a dataset sampled from the mixture distribution $\frac{1}{2}(\rho^E(s,a) + \rho_{\text{RL}}(s,a))$, where $\rho^E(s,a)$ is the distribution of data under the expert. Since the rewards are one for the data sampled from $\rho^E$ and zero for state-action pairs sampled from $\rho_{\text{RL}}$, the expected reward obtained at a state-action pair $(s,a)$ is given by $\frac{\rho^E(s,a)}{\rho^E(s,a)+\rho_{\text{RL}}(s,a)}$, which is non-stationary and varies between zero and one. The benefit of SQIL is that it does not require a support estimate. However, while SQIL has demonstrated good empirical performance, it does not come with a theoretical guarantee of any kind, making it hard to deploy in settings where we need a theoretical certificate of policy quality.

**Expert Feedback Loop**   IL algorithms such as SMILe (Ross & Bagnell, 2010) and DAGGER (Ross et al., 2011) assume the ability to query the expert for more data. While this makes it easier to reproduce expert behavior, the ability to execute queries is not always available in realistic scenarios. Our reduction is one-off, and does not need to execute expert queries to obtain more data.

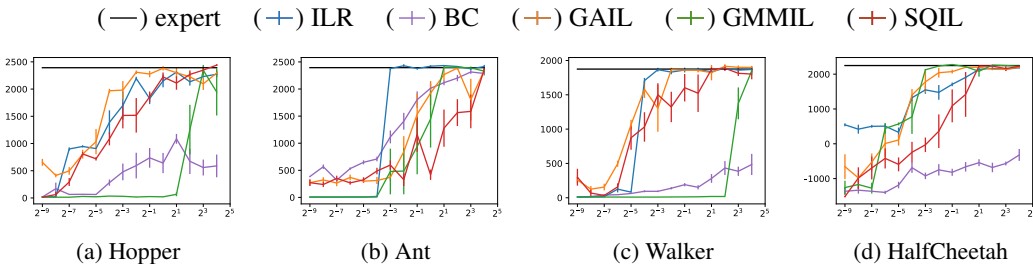

Figure 1: Extrinsic return of imitation learners as a function of the number of episodes of demonstration data. For comparison, we also show expert performance. Confidence bars denote one standard error.

## 6 EXPERIMENTS

We investigate the performance of our reduction on continuous control tasks. Because Wang et al. (2019) have already conducted an extensive evaluation of various ways of estimating the support for the expert distribution, we do not attempt this. Instead, we focus on two aspects: the amount of expert data needed to achieve good imitation and the relationship between intrinsic and extrinsic return.

**Continuous Relaxation** In order to adapt the discrete reduction to a continuous state-action space, we define the reward as $R'_I(s, a) = 1 - \min_{(s', a') \in D} d_{L_2}((s, a), (s', a'))^2$. To see how this is related to the reward in equation 6, consider a scaled version of the binary reward used in the theoretical analysis which ranges from $1 - L$ to 1 instead of from 0 to 1, where $L$ is the $L_2$-diameter of the state-action space. Then the relaxed reward is an upper bound on the scaled theoretical reward. Since optimizing this upper bound is different from optimizing the reward in equation 6 directly, we can no longer theoretically claim the recovery of expert reward (as in Lemma 7). However, this property still holds in practice, as verified below. In settings where such reduction is not feasible, the random expert distillation approach can be used, which Wang et al. (2019) have demonstrated can work quite well and Burda et al. (2018) have shown can work with pixel observations.

**Experimental Setup** We use the Hopper, Ant, Walker and HalfCheetah continuous control environments from the `PyBullet` gym suite (Ellenberger, 2018–2019). The expert policy is obtained by training a SAC agent for 2 million steps. We compare 5 methods. ILR denotes our imitation learning reduction. BC denotes a behavioral cloning policy, obtained after training on the expert dataset for 500000 steps. GAIL (Ho & Ermon, 2016) denotes the Generative Adversarial Imitation Learning algorithm. GMMIL (Kim & Park, 2018) denotes a variant of adversarial IL minimizing the MMD divergence. SQIL (Reddy et al., 2020) denotes the SQIL heuristic. In order to ensure a fair comparison among algorithms, we re-implemented all of them using the same basis RL algorithm, SAC (Haarnoja et al., 2018). All imitation learning agents were run for 500000 environment interactions. We repeated our experiments using 5 different random seeds. The remaining details about the implementation and hyperparameters are provided in Appendix A.

**Performance as Function of Quantity of Expert Data** The plot in Figure 1 shows the performance of various imitation learners as a function of how much expert data is available. The amount of data is measured in episodes, where fractional numbers mean that state-action tuples are taken from the beginning of the episode. Confidence bars represent one standard error. All of the methods except BC work well when given a large quantity of expert data (16 episodes, on the right hand side of the plot). However, for smaller sizes of the expert dataset, the performance of various methods deteriorates at different rates. In particular, GMMIL needs more expert data than other methods on Hopper and Walker, while on Ant ILR achieves top performance with less data than other methods. Across all the environments, ILR is competitive with the best of the other methods. We claim that our method is preferable since it is the simplest one to implement, is computationally cheap and does not introduce new hyperparameters beyond those of the RL solver.

**Performance as Function of Quantity of Environment Interactions** In Figure 2, we examine the progress of the various agents as a function of environment interactions for a dataset containing 16 expert episodes. ILR shows improved sample efficiency on Ant, while behaving similarly to other

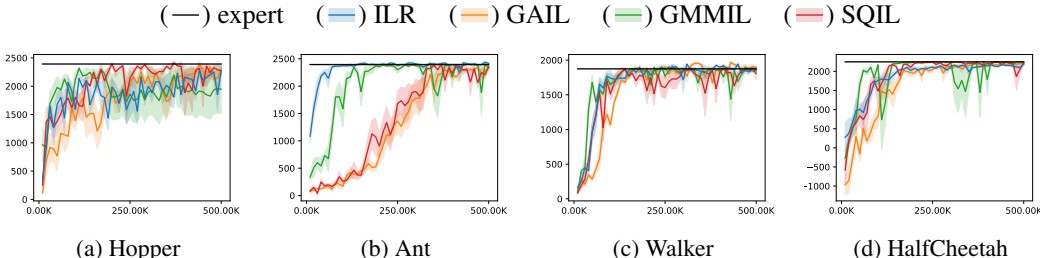

| (a) Hopper | (b) Ant | (c) Walker | (d) HalfCheetah |

Figure 2: Extrinsic return of imitation learner as a function of the number of environment interactions. Imitation learners were trained on 16 expert trajectories. Shaded area denotes one standard error.

algorithms on Hopper, Walker and HalfCheetah. Again, we are led to conclude that ILR is preferable to to the other algorithms because of its simplicity. We provide analogous plots for different sizes of the expert dataset in Appendix B.

**Intrinsic and Extrinsic Returns are Related in Practice** Our reasoning in Section 4.3 shows that achieving a large intrinsic return serves as a performance certificate, guaranteeing a certain level of extrinsic return. We conducted an experiment to determine if this property still holds for our continuous-state relaxation. Figure 3 shows the relationship between the intrinsic return (on the horizontal axis) and the extrinsic return (on the vertical axis), during the first 50K steps of training using the Hopper environment. Each point on the plot corresponds to evaluating the policy 100 times and averaging the results. The plot confirms that there is a clear, close-to-linear, relationship between the intrinsic and extrinsic return.

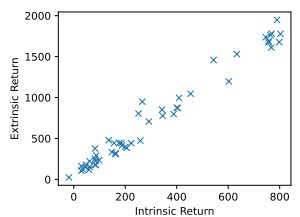

Figure 3: Intrinsic vs Extrinsic returns

## 7 CONCLUSIONS

We have shown that, for deterministic experts, imitation learning can be performed with a single invocation of an RL solver. We have derived a bound guaranteeing the performance of the obtained policy and relating the reduction to adversarial imitation learning algorithms. Finally, we have evaluated the proposed reduction on a family of continuous control tasks, showing that it achieves competitive performance while being comparatively simple to implement.

## 8 ACKNOWLEDGMENTS

The author thanks Lucas Maystre and Daniel Russo for proofreading the draft and making helpful suggestions. Any mistakes are my own.

## 9 ETHICS STATEMENT

Since our most important contribution is a bound stating that the imitation learner closely mimics the expert, the ethical implications of actions taken by our algorithm crucially depend on the provided demonstrations. The most direct avenue for misuse is for malicious actors to intentionally demonstrate unethical behavior. Second, assuming the demonstrations are provided in good faith, there is a risk of excessive reliance on the provided bound. Specifically, it is still possible that our algorithm fails to recover expert behavior in situations where assumptions needed by our proof are not met, for example if the expert policy is stochastic. In certain settings, the bounds can also turn out to be very loose, for example when the expert policy takes long to mix. This means that one should not deploy our reduction in safety-critical environments without validating the assumptions first. However, the proposed reduction also has positive effects. Specifically, we provide the benefits of adversarial imitation learning, but without having to train the discriminator. This means that training is cheaper, making imitation learning more affordable and research on imitation learning more democratic.

## 10 REPRODUCIBILITY STATEMENT

Our main contribution is Proposition 1, for which we provided a complete proof. The external results we rely on are generic properties of the Total Variation distance and of Markov chain mixing[1], for all of which we have provided references. Our experimental setup is described in detail in Appendix A. Moreover, we make the source code available.

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

## A  DETAILS OF EXPERIMENTAL SETUP

To ensure a fair comparison, we implemented all imitation learners (ILR, GAIL, GMMIL and SQIL) using the same base reinforcement learning solver (SAC). The implementation of SAC we used as a basis came from the d4rl-pybullet repository[2]. The hyperparameters of SAC are given in the table below.

| hyperparameter | value |
|---|---|
| actor optimizer | Adam |
| actor learning rate | 3e-4 |
| critic optimizer | Adam |
| critic learning rate | 3e-4 |
| batch size | 100 |
| update rate for target network (tau) | 0.005 |
| $\gamma$ | 0.99 |

The policy network and the critic network have two fully connected middle layers with 256 neurons each followed by ReLU non-linearities. The entropy term was not used when performing the actor update after we found that including it had no effect on performance. Instead, a fixed Gaussian noise with standard deviation of 0.01 was used for exploration.

For our behavioral cloning baseline, we performed supervised learning for 500K steps, using the Adam optimizer with learning rate 3e-4. We used the same policy network architecture as the SAC policy network.

ILR does not introduce any additional hyperparameters.

GAIL has the following additional hyperparameters.

| hyperparameter | value |
|---|---|
| discriminator optimizer | Adam |
| discriminator learning rate | 3e-4 |
| discriminator batch size | 100 |
| discriminator update frequency | 32 |
| discriminator training iterations | 500 |

---

[2]https://github.com/takuseno/d4rl-pybullet

The last two hyperparameters mean that the discriminator network in GAIL is updated every 32 simulation steps, using 500 batches of data.

A direct implementation of GMMIL requires looping over the whole replay buffer each time a reward is computed, which makes the algorithm very slow for longer runs. To make it tractable, we we only computed the kernel for 8 points closest to the given state-action pair in the replay buffer and for 8 points in the expert dataset. The data structure used to find the nearest 8 points was updated every 32 steps. Kernel hyper-parameters were automatically estimated using the median heuristic (Kim & Park, 2018) each time the algorithm was run.

SQIL uses a batch size of 200 (100 for state-action pairs from the expert and 100 for state-action pairs coming from the replay buffer). It does not have additional hyperparameters.

## B  PLOTS MEASURING SAMPLE EFFICIENCY

In this section, we include plots showing the efficiency of all algorithms measured in terms of environment interactions, for different quantities of available expert data. The plots can be thought as an extension of Figure 1 in the main paper to what is happening during learning, not just at the end.

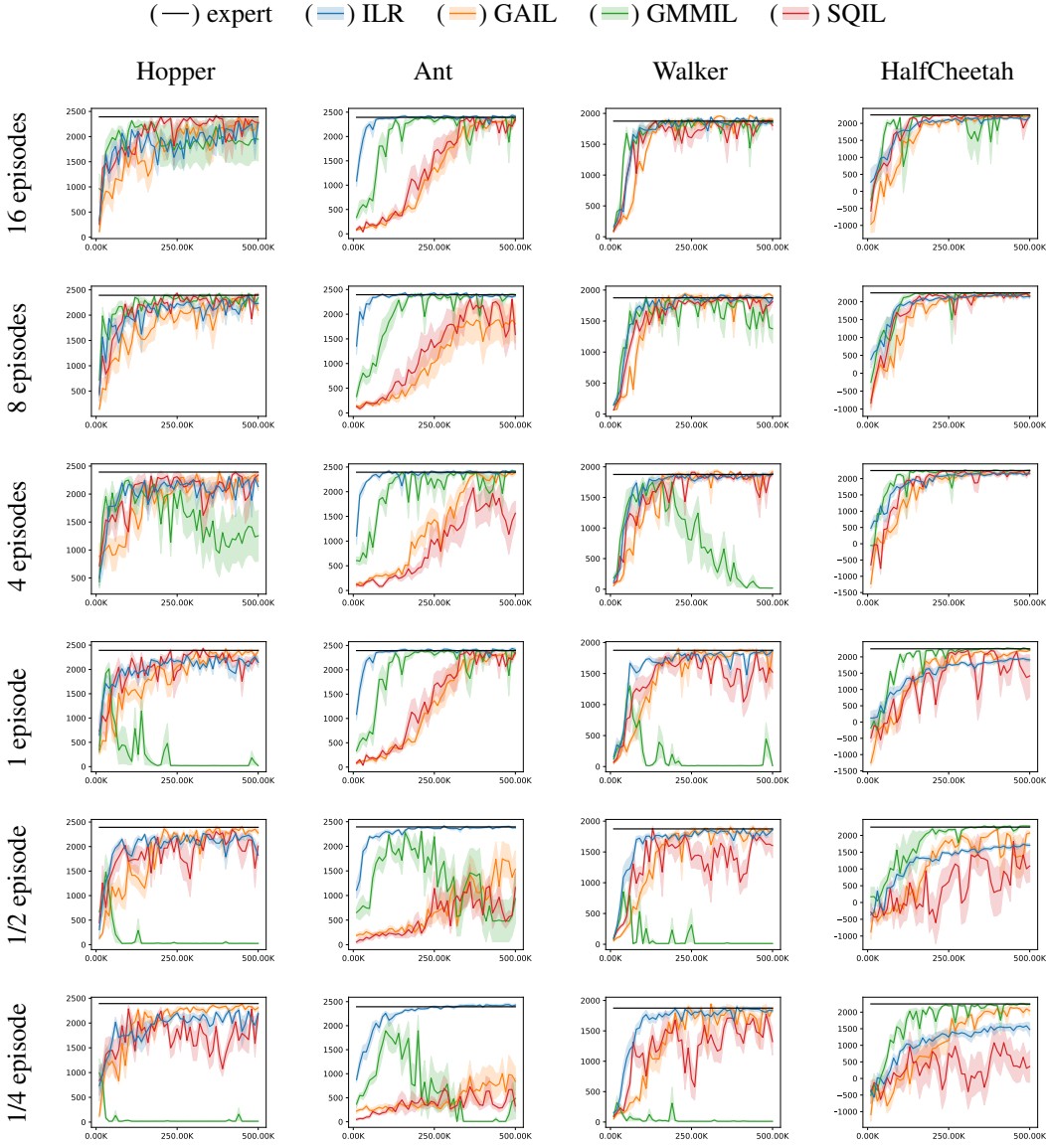

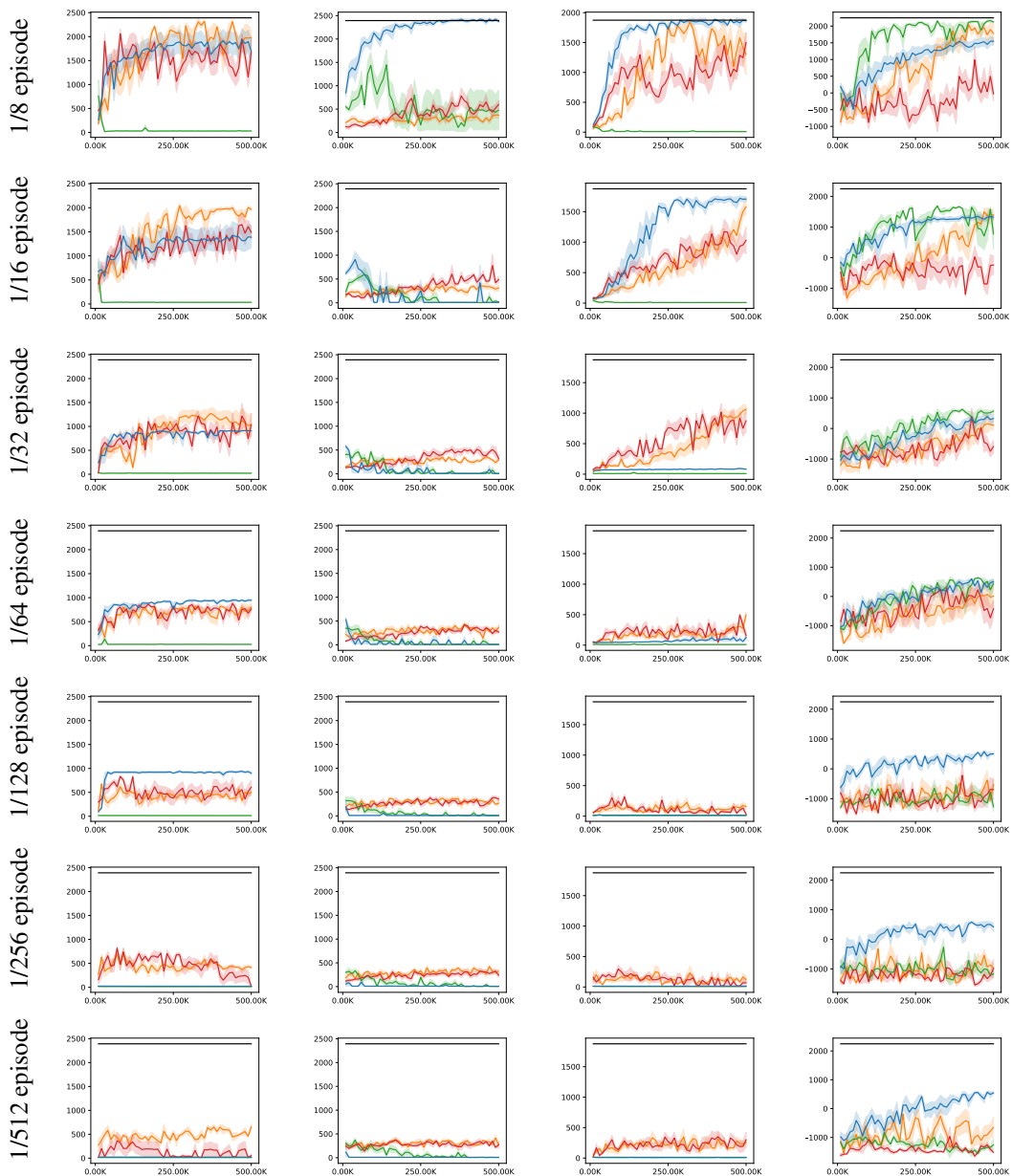

## C    RUNTIME COMPARISON OF DIFFERENT ALGORITHMS

We ran all algorithms from the paper in a controlled computational setting (no other processes running on the machine) in the hopper environment for 500K iterations. An expert dataset containing 16 episodes was used. Results are given below.

| Algorithm | Runtime |
|-----------|---------|
| BC | 0m 17s |
| SQIL | 94m 39s |
| ILR | 99m 16s |
| GAIL | 213m 56s |
| GMMIL | 448m 42s |

Behavioral Cloning (BC) is the fastest, because it does not have to do any interaction with the environment (it does, however, produce a policy with the worst performance). SQIL and ILR have a

comparable runtime, dominated by the complexity of simulating the environment. GAIL is slower because it has to train the discriminator network. GMMIL is the slowest algorithm due to the high cost of computing the MMD divergences.

Overall, the conclusion from this benchmark is that our algorithm (ILR) is highly competitive in terms of runtime.

# D ADDITIONAL PROOFS

*Proof of Lemma 1.* By setting $v' = 1 - v$, it follows that $\mathbb{E}_{\rho_2}[v] - \mathbb{E}_{\rho_1}[v] \leq \epsilon$ for all $v$. Therefore $|\mathbb{E}_{\rho_1}[v] - \mathbb{E}_{\rho_2}[v]| \leq \epsilon$ for all $v$. For any $M \subseteq X$ on the right-hand side of equation 3, we can instantiate $v(x) = \mathbb{1}\{x \in M\}$, completing the proof. □

*Proof of Lemma 2.* We have

$$|\mathbb{E}_{\rho_1}[v] - \mathbb{E}_{\rho_2}[v]| = |\textstyle\sum_x (\rho_1(x) - \rho_2(x))v(x)| \leq \textstyle\sum_x |\rho_1(x) - \rho_2(x)| = \|\rho_1 - \rho_2\|_1.$$

Here, the first inequality follows because elements of $v$ are in the interval $[0, 1]$. The statement of the lemma follows from the property $\|\rho_1 - \rho_2\|_{\text{TV}} = \frac{1}{2}\|\rho_1 - \rho_2\|_1$. □

*Proof of Lemma 3.* By Theorem 4.9 in the book by Levin et al. (2017), there are constants $C > 0$ and $\alpha \in (0, 1)$ such that $\max'_s \|\mathbb{1}(s')^\top P_E^t - \rho_S^E\|_{\text{TV}} \leq C\alpha^t$. Defining $\tilde{\rho}_t^{s'}(s, a) = (\mathbb{1}(s')^\top P_E^t)(s)\pi_E(a|s)$, we have $\max_{s'} \|\tilde{\rho}_t^{s'} - \rho^E\|_{\text{TV}} \leq C\alpha^t$, where we used the fact that the expert policy is deterministic so that $\|\tilde{\rho}_t^{s'} - \rho^E\|_{\text{TV}} = \|\mathbb{1}(s')^\top P_E^t - \rho_S^E\|_{\text{TV}}$. The mixing time of the chain $\tau_{\text{mix}}$, defined as the smallest $t$ so that $\max_{s'} \|\mathbb{1}(s')^\top P_E^t - \rho_S^E\|_{\text{TV}} \leq \frac{1}{4}$ is then bounded as $\tau_{\text{mix}} \leq -\log_\alpha C - \log_\alpha(4)$.

It remains to show that, for any probability distribution $p_1$, we have $\sum_{t=0}^\infty \|p_1^\top P_E^t - \rho_S^E\| \leq 2\tau_{\text{mix}}$.

Define $d(t) = \sup_p \|p^\top P_E^t - \rho_S^E\|_{\text{TV}}$. By equation 4.35 in the book by Levin et al. (2017), we have

$$d(\ell\tau_{\text{mix}}) \leq 2^{-\ell}. \tag{20}$$

We can now write

$$\sum_{t=0}^\infty \|p_1^\top P_E^t - \rho_S^E\| \leq \sum_{t=0}^\infty d(t) \leq \sum_{t=0}^\infty d(\tau_{\text{mix}}\lfloor t/\tau_{\text{mix}}\rfloor) \leq \sum_{t=0}^\infty 2^{-\lfloor t/\tau_{\text{mix}}\rfloor},$$

where the first inequality follows from the definition of $d(t)$, the second one from the fact that $d(t') \leq d(t)$ for $t' \geq t$ and the last one from equation 20. We can rewrite the right-hand side further, giving

$$\sum_{t=0}^\infty 2^{-\lfloor t/\tau_{\text{mix}}\rfloor} = \tau_{\text{mix}} \sum_{t=0}^\infty 2^{-t} = 2\tau_{\text{mix}}.$$

□

*Proof of Lemma 7.* Consider the imitation learner's trajectory of length $T$. We will now consider sub-sequences where the agent agrees with the expert. Denote by $M_\ell$ the number of such sequences of length $\ell$. Denote by $\hat{V}^{i,\ell}$ the per-step extrinsic reward obtained by the agent in the $i$th sequence of length $\ell$.

Assuming the worst case reward on states which are not in any of the sequences (i.e. where the agent disagrees with the expert), the total extrinsic return $J_T$ along the trajectory is at least $\sum_\ell \ell \sum_{i=1}^{M_\ell} \hat{V}^{i,\ell} = \sum_\ell \ell M_\ell \frac{1}{M_\ell} \sum_{i=1}^{M_\ell} \hat{V}^{i,\ell}$. Dividing by the sequence length, the per-step extrinsic reward is at least

$$\frac{J_T}{T} \geq \sum_\ell \frac{\ell M_\ell}{T} \frac{1}{M_\ell} \sum_{i=1}^{M_\ell} \hat{V}^{i,\ell}. \tag{21}$$

Denote by $\Delta_\ell = \left| \frac{1}{M_\ell} \sum_{i=1}^{M_\ell} \hat{V}^{i,\ell} - \frac{1}{M_\ell} \sum_{i=1}^{M_\ell} \tilde{V}^\ell_{p(s_1^\ell)} \right|$ the difference between the average extrinsic reward obtained in sequences of length $\ell$ and its expected value, where we denoted by $p(s_1^\ell)$ the initial state distribution among sequences of length $\ell$. We can now re-write equation 21 as:

$$\frac{J_T}{T} \geq \sum_\ell \frac{\ell M_\ell}{T} \frac{1}{M_\ell} \sum_{i=1}^{M_\ell} \tilde{V}^\ell_{p(s_1^\ell)} - \sum_\ell \frac{\ell M_\ell}{T} \Delta_\ell. \tag{22}$$

Using Lemma 6, we can re-write this further

$$\frac{J_T}{T} \geq \sum_\ell \frac{\ell M_\ell}{T} \left( \mathbb{E}_{\rho^E}[R] - \frac{1}{\ell} 4\tau_{\text{mix}} \right) - \sum_\ell \frac{\ell M_\ell}{T} \Delta_\ell \tag{23}$$

$$= \sum_\ell \frac{\ell M_\ell}{T} \mathbb{E}_{\rho^E}[R] - \sum_\ell \frac{M_\ell}{T} 4\tau_{\text{mix}} - \sum_\ell \frac{\ell M_l}{T} \Delta_\ell. \tag{24}$$

Denote by $B$ the number of timesteps the imitation learner disagrees with the expert. Observe that we have $\sum_\ell \frac{M_\ell}{T} \leq \frac{B+1}{T}$ since $B$ bad timesteps can partition the trajectory into at most $B + 1$ sub-sequences and $\sum_\ell M_\ell$ is the total number of sub-sequences. This gives

$$\frac{J_T}{T} \geq \sum_\ell \frac{\ell M_\ell}{T} \mathbb{E}_{\rho^E}[R] - \frac{B+1}{T} 4\tau_{\text{mix}} - \sum_l \frac{\ell M_\ell}{T} \Delta_\ell. \tag{25}$$

Now, using the fact that the imitation learner policy is ergodic, we can take limits as $T \to \infty$. The left hand side converges to $\mathbb{E}_{\rho^I}[R]$ with probability one. On the right-hand side, $\sum_\ell \frac{\ell M_\ell}{T}$ (the fraction of time the imitation learner agrees with the expert) converges to $\mathbb{E}_{\rho^I}[R_{\text{int}}] = 1 - \kappa$ with probability one. Using ergodicity again, $\frac{B+1}{T}$ converges to $1 - \mathbb{E}_{\rho^I}[R_{\text{int}}]$ with probability one.

It remains to prove that the error term $\sum_\ell \frac{\ell M_\ell}{T} \Delta_\ell$ converges to zero with probability one. First, we will show that, for every $\ell$, either $M_\ell$ is always zero, i.e. the streak length $\ell$ does not occur in sub-sequences of any length or $M_\ell \to \infty$ as $T \to \infty$. Indeed, the chain is ergodic, which means that, if we traversed a streak of length $\ell$, we have non-zero probability of returning to where the streak begun and then retracing it. Let us call the of set of all $\ell$s where $M_\ell = 0$ by $\mathbb{N} - L_\infty$. Moreover, let us call the set of $\ell$s with $M_\ell \to \infty$ as $T \to \infty$ with $L_\infty$. We have:

$$\sum_{\ell \in L_\infty} \frac{\ell M_\ell}{T} \Delta_\ell + \sum_{\ell \in \mathbb{N} - L_\infty} \frac{\ell M_\ell}{T} \Delta_\ell \leq \sup_{\ell \in L_\infty} \Delta_\ell. \tag{26}$$

The second term in the sum equals zero with because of the definition of $L_\infty$. The term $\sup_{\ell \in L_\infty} \Delta_\ell$ converges to zero with probability one by the law of large numbers, where we use the fact that $M_\ell \to \infty$ as $T \to \infty$ for $M_\ell \in L_\infty$. $\square$

