# OpenReview forum: "Imitation Learning by Reinforcement Learning"
_ICLR.cc/2022/Conference — ICLR 2022 Poster_

### Official Review · Reviewer_VY5r · 2021-10-31

**Correctness:** 3
**Technical Novelty And Significance:** 3
**Empirical Novelty And Significance:** 2
**Recommendation:** 6
**Confidence:** 4

**Main Review:**

Strength:
1. The paper shows that imitation learning (IL) may be reduced to RL with stationary reward for deterministic experts. The analysis provides the theoretical grounding for several existing IL methods. The paper also contributes the connection between the stationary reward to minimizing f-divergence typically seen in adversarial IL, proving that the stationary reward minimizes total variation distance.

2. The proof is clear, well motivated and easy to follow.

3. The experiment studies an important aspect of IL, which is the relationship between the amount of expert data and the IL's performance.

Weakness:
1. Similar to previous works, the practical algorithm uses soft reward (1 - min L_2 distance) to avoid reward sparsity. While a very reasonable choice, the soft reward violates the critical assumption of deterministic expert. Could the authors explain the implications of the soft reward on the theoretical results?

2. The empirical evaluation is limited, with several related questions unaddressed. For instance, the SAC expert appears to be rather weak (in terms of its performance) across different environments. It is unclear whether similar patterns would be observed if the expert matches the level of the performance observed in previous works (e.g. https://github.com/berkeleydeeprlcourse/homework/tree/master/hw1/experts). Another question is how the expert is trained has any impact on the IL performance. For instance, SAC implementation typically directly constraint the range of the action, while a Gaussian expert only clips the action as a post-processing step. In my experiences, these details have quite significant impacts on IL's performance and I hope the authors could expand on the experiments to further validate their theoretical results.

Questions:
1. Comparison between Wang et al. 2019 vs the 1-min L_2 distance reward. Could the authors motivate why the latter reward is used and how it compares to Wang et al. 2019?

2. Could the authors further elaborate on the assumption of the deterministic expert? E.g. How much does it matter in the practical algorithm? Is it even possible to differentiate between stochastic vs deterministic one from limited training data?

**Summary Of The Paper:**

The paper contributes a theoretical analysis of imitation learning (IL) under deterministic experts. The paper shows that IL in this setting can be reduced to RL with a stationary reward, and the stationary reward minimizes the total variation distance between the expert and the learner. Several empirical experiments were presented to validate the theoretical analysis.

**Summary Of The Review:**

The paper shows that, for deterministic experts, imitation learning (IL) could be reduced to RL with a stationary reward. The paper thus provides theoretical grounding for several existing works. In addition, the paper also draws connection with f-divergence minimization commonly found in adversarial IL. On the other hand, the empirical results are consistent with those from existing works, but are limited in its scope. Despite the flaws, I think the work is still interesting to the community. It would be nice if the paper could include more experiments to further support the theoretical analysis.

---

> ### Author Response · Authors · 2021-11-14
> **Thank you for the review!**
>
> We would like to first address the weaknesses you point out in your review.
>
> Concerning weakness 1, we agree that using the soft reward requires more explanation. To see how the soft reward and the theoretically-motivated reward are related, consider a scaled version of the binary reward used in the theoretical analysis which ranges from -L to 1 instead of from 0 to 1 (where L is the L2-diameter of the state-action space). The relaxed reward is an upper bound on the theoretical reward. While we can no longer theoretically claim that optimizing this upper bound guarantees the recovery of expert reward (as in Lemma 4), this property still holds in practice, as verified by the experiment reported in the "Intrinsic and Extrinsic Returns are Related in Practice" paragraph. We agree that this justification should have been included and will do so in the updated version of the paper. However, we disagree with your suggestion that using a soft reward violates the assumption that the expert is deterministic. In fact, our experiments use only deterministic expert trajectories, as required by the theory.
>
> Concerning weakness 2, thank you for pointing us to a repository with expert policies. However, these policies are meant for the MuJoCo simulator, while we used PyBullet, the only open-source choice available at the time our paper was being written. Because the PyBullet and MuJoCo environments are different, the scores will be different as well. We are not aware of existing works using the PyBullet benchmarks showing performance substantially better than what we had in our paper. Concerning your question about how the action range is constrained, we  used the tanh non-linearity (as is common with SAC). We would be glad to run additional ablations if you have some concrete ones you could suggest.
>
> We now wanted to address your questions.
>
> Concerning question 1, we introduced the 1-L2 distance reward as a simple relaxation of the theoretical reward in cases where the L2 distance is available (see also our explanation of the soft reward above). In cases where the L2 distance is unavailable, the expert distillation reward used by Wang et al. (2019) can be used.
>
> Concerning question 2, the assumption of a deterministic expert is crucial for our theoretical analysis. It is possible to come up with grid-world examples where the algorithm does not converge to a policy close to the expert when provided with samples from a stochastic expert. However, since for any MDP it is possible to find an optimal policy which is deterministic, this should not be a major problem if the goal is to maximize extrinsic return. Moreover, in many practical settings, it is natural to provide demonstrations from a deterministic expert. It is certainly true that in the continuous control tasks, the expert policies are naturally deterministic (SAC and other existing algorithms are normally evaluated using a deterministic policy). We did not examine the question of how to determine whether or not an expert is deterministic from training data alone. We consider that an interesting, but separate, research direction. Instead, we address the setting where we can trust the expert to provide a rollout from a deterministic policy.

---

> > ### Comment · Reviewer_VY5r · 2021-11-18
> > **Thanks for the clarification**
> >
> > I thank the authors for the further explanation.
> >
> > For the connection between theoretical results and the experiments, I'm not entirely convinced on the link between intrinsic and extrinsic returns from the empirical validation. From my experiences, some high intrinsic returns could correspond to degenerate solutions in the extrinsic reward space, depending on which RL algorithm is used for imitation learning, as well as the quality and size of the demonstration data.
> >
> > I agree with the other reviewers that the Pybullet experiments may not be entirely sufficient to validates the theoretical results. It would be good to try out the Mujoco environment (and the humanoid task from Mujoco suggested by another reviewer).
> >
> > I would keep my current score because I believe the theoretical contributions are of interest to the community, but the empirical evaluations could be improved.

---

> > > ### Author Response · Authors · 2021-11-19
> > > **Thanks for the comment.**
> > >
> > > You say that you are concerned that a high intrinsic return could correspond to bad solutions in the extrinsic reward space. We wanted to point out that in all our benchmarks, we evaluate the extrinsic return.  Empirically, our method (ILR) achieves good performance as measured by the extrinsic return.
> > >
> > > Theoretically (for finite MDPs), the link between intrinsic and extrinsic returns is provided in Lemma 4. This link does not depend on the algorithm being used to do RL.

---

> > > > ### Comment · Reviewer_VY5r · 2021-11-20
> > > > **Further clarification**
> > > >
> > > > I thank the authors for the further input.
> > > >
> > > > As highlighted by other reviewers and me, the empirical evaluation is a significant relaxation of the theoretical results. The reward is dense in the empirical evaluation and there is no guarantee that the theorem could apply. In addition, as highlighted by reviewer TFjU, the theorem also requires a large amount of demonstration proportional to $|S|^2$ in the worst case. In the continuous-space environments tested in the paper, could the authors elaborate on if $|S|^2$ is still bounded?
> > > >
> > > > My comments on the link between extrinsic and intrinsic reward was based on similar experiments that I performed in the past using Mujoco environments. Given the relaxation of theoretical assumption in the empirical evaluation, more thorough experiments considering various settings would be more convincing in validating the theoretical claims.

---

> > > > > ### Author Response · Authors · 2021-11-21
> > > > > **Thanks for the further clarification.**
> > > > >
> > > > > You are right that the reward used in the empirical evaluation is a relaxation of the original binary reward. Formally, our theoretical result holds for finite MDPs and cannot be applied to MDPs with a continuous state space (which we state in the paper).
> > > > >
> > > > > You refer to the link between the extrinsic and intrinsic rewards and the experiments you performed in the past, which sometimes resulted in degenerate solutions. We find it difficult to respond to these experiments since you do not provide a citation. Would it be possible for you to provide a link to an anonymous code repository so we can explain the difference?
> > > > >
> > > > > You also refer to the need for more thorough experiments. Aside from using MuJoCo, do you have concrete suggestions for new experiments you feel are needed?

---

> > > > > > ### Comment · Reviewer_VY5r · 2021-11-22
> > > > > > **Clarification**
> > > > > >
> > > > > > For the correlation between intrinsic and extrinsic rewards, please see http://proceedings.mlr.press/v139/hussenot21a/hussenot21a.pdf, which uses intrinsic rewards as a proxy of extrinsic rewards for model selection. Wang et al. 19 was used as one of the intrinsic reward. In figure 2, it is quite clear that the correlation between intrinsic and extrinsic reward depends on the IL algorithm and the test environment.
> > > > > >
> > > > > > I don't understand the apparent unwillingness to incorporate Mujoco experiments (which is now free to use) during the discussion phase. Expert policies may be obtained easily from open-source projects and the proposed algorithm should require minimum changes for the new environments.

---

> > > > > > > ### Author Response · Authors · 2021-11-22
> > > > > > > **Re: Clarification**
> > > > > > >
> > > > > > > Thanks for the clarification and for pointing us to Hussenot, Andrychowicz, Vincent et al. Our Figure 3 supports the conclusion that the intrinsic and extrinsic returns are closely related in our case.
> > > > > > >
> > > > > > > Concerning your point about reimplementing the experiments in MuJoCo: unfortunately we had neither the time for reimplementation nor for re-running (running the experiments for all seeds and dataset sizes takes several days). We think that the PyBullet experiments are enough to support our main points.

---

### Official Review · Reviewer_7ezW · 2021-10-31

**Correctness:** 4
**Technical Novelty And Significance:** 2
**Empirical Novelty And Significance:** 2
**Recommendation:** 6
**Confidence:** 3

**Main Review:**

The paper includes potentially interesting results. Some comments that can improve the paper are given below.
1) In this paper, average rewards are considered. Is there any reason to consider the average reward MDP instead of discounted MDP, which is more widely used?
2) It is not clear what the norm || ||_{TV} means.
3) It would be better to discuss what kinds of RLs can be used in Proposition 1. It is not clear in the current paper.
4) It would be better if the definition of intrinsic reward and extrinsic reward is explained in this paper.
5) It is not clear how the imitation learner can learn the expert policy from the intrinsic reward in section 3.
It would be better if the process is explained in more details.
6) In the experiment, it is not clear how the comaprative analysis is done. Because the authors did not develop new algorithms,
it would be better to explain details of the comparative analysis.
 7) Overall, the organization of the paper can be improved further to more clearly deliver the ideas by adding more detailed backgrounds.

**Summary Of The Paper:**

In this paper, the authors show that, for deterministic experts, imitation learning can be done by reduction
to reinforcement learning with a stationary reward. theoretical analysis both
certifies the recovery of expert reward and bounds the total variation distance
between the expert and the imitation learner, showing a link to adversarial imitation
learning. Experiments are given to confirm that the reduction works well in
practice for continuous control tasks.


**Summary Of The Review:**

In sum, the paper seems to contain potentially interesting results.
The overall organization and presentation can be further improved to improve the readability of the paper.

---

> ### Author Response · Authors · 2021-11-14
> **Thank you for the review!**
>
> 1. We decided to study the average-reward formulation because it preserves the essential features of the imitation learning problem, while being conceptually simpler. However, we believe that our analysis could be extended to the discounted MDP case.
> 2. In the original submission, we defined the total variation norm in appendix C. We agree that the definition should have been placed in the main paper. We will prepare an updated version of the paper with the definition moved and more prominent.
> 3. Our algorithm can work with any RL solver. We use SAC (soft actor-critic) in our experiments, but in principle any algorithm can be used.
> 4. We define intrinsic reward in section 3. Our Proposition 1 holds for any extrinsic reward signal. We will attempt to introduce these concepts more clearly in the revised version of the paper.
> 5. The RL agent learns a policy by interacting with the environment and learning using the intrinsic reward signal defined in section 3. In principle, this process is the same as a vanilla RL training loop.
> 6. We explain the details of our comparison (hyperparameters etc) in appendices A and B. If there is anything specific you find unclear, please follow up with a message and we will answer.
> 7. We will re-organize the paper, putting the content of appendices C and D into the main body and providing the background about the total variation distance earlier on in the paper.

---

### Official Review · Reviewer_XK3g · 2021-11-02

**Correctness:** 3
**Technical Novelty And Significance:** 3
**Empirical Novelty And Significance:** 3
**Recommendation:** 6
**Confidence:** 4

**Main Review:**

The idea of using reinforcement learning to imitate expert behavior is simple to execute compared to other imitation learning methods. I have a concern regarding the sparse reward function used in this paper. When training a reinforcement learning model, the sparse reward becomes problematic when you apply RL algorithm to solve it. Although the better intrinsic reward leads to better extrinsic reward, it is still troublesome to get a good policy by any reinforcement learning algorithm with this sparse reward function. This makes the result less useful than just applying other comparison methods. Could you give more explanation on this?
In term of computation time, the sparse reward can cause problem of bad convergence rate. It is much clearer if you can provide comparison of computation time of different algorithms in the experiment section.
Overall, this paper is well written and very clear to readers. The result is promising and interesting from a theoretical point of view.


**Summary Of The Paper:**

This paper considers imitation learning problem which aims at obtaining a policy that imitate expert behavior. Authors considers using reinforcement learning with a stationary reward constructed by expert datasets. Through theoretical analysis, the corresponding imitation policy is proved to achieve high expected per-step intrinsic reward and extrinsic reward. The difference between the expert policy and the imitation policy is bounded with a high probability. Some empirical experiments are performed on continuous control tasks. The results show this method is comparable with other algorithms while the algorithm is simpler.

**Summary Of The Review:**

This paper provides rigorous analysis for using reinforcement learning in imitation learning. The result is novel and shows the potential to accomplish imitation learning in a straightforward way. The intrinsic reward function reduced by the expert dataset is simple to construct. By setting a constant one reward for matching the demonstrated action in a demonstrated state, any reinforcement learning algorithm can provide policy for such a MDP problem. The result in this paper gives intuition of how this method performs in terms of expected return. And sufficient experimental results are shown to verify their theoretical results.

---

> ### Author Response · Authors · 2021-11-14
> **Thank you for the review!**
>
> You mention that the sparse reward signal may make it difficult for the agent to learn a good imitation policy. This is true. However, this can be mitigated in two ways. First, in practical applications with large or infinite state spaces, support estimators returning non-binary values can be used, akin to the ones introduced in equation (8) of Wang et al. (2019). Second, off-policy RL solvers can be used which eliminate the requirement that the algorithm find the state-action pairs with reward one by exploration. In other words, imitation learning with a sparse reward signal is very different from a standard RL task with sparse reward because in imitation learning it is known which state-action pairs are associated with a reward and, unlike in standard RL, the agent does not have to find them.
>
> You also asked for a comparison of computation time of different algorithms. We will run them again in a controlled runtime environment and report the numbers here. We will also update the paper with the benchmark.

---

> > ### Author Response · Authors · 2021-11-18
> > **Runtime benchmark.**
> >
> > As requested in the review, we have run all algorithms from the paper in a controlled computational environment (no other processes running on the machine). We ran the imitation learning algorithms in the hopper environment for 500K iterations. An expert dataset containing 16 episodes was used. Results are given below.
> >
> > | Algorithm      | Runtime  |
> > | ----------- | ----------- |
> > | BC      | 0m 17s       |
> > | SQIL      | 94m 39s       |
> > | ILR (ours)      | 99m 16s       |
> > | GAIL     | 213m 56s       |
> > | GMMIL     | 448m 42s       |
> >
> > Behavioral Cloning (BC) is the fastest, because it does not have to do any interaction with the environment (it does, however, produce a policy with the worst performance). SQIL and ILR have a comparable runtime, dominated by the complexity of simulating the environment. GAIL is slower because it has to train the discriminator network. GMMIL is the slowest algorithm due to the high cost of computing the MMD divergences.
> >
> > Overall, the conclusion from this benchmark is that our algorithm is highly competitive in terms of runtime.

---

### Official Review · Reviewer_TFjU · 2021-11-02

**Correctness:** 4
**Technical Novelty And Significance:** 3
**Empirical Novelty And Significance:** 2
**Recommendation:** 5
**Confidence:** 4

**Main Review:**

The paper is generally well written and easy to follow; however, the structure is sometimes unusual. The proofs would be easier to follow if the authors moved Lemmas from Appendix C & D into the main body while moving some of the detailed proofs from the main body into the Appendix.

The derived bound is non-trivial, but it is unclear how significant it is. The main issue lies in the assumption on the amount of data that is required which is linear (with a factor > 1) in the squared size of the state-space (note that the bound remains large even if \delta is set to 1). Such a large amount of expert data is required since rewarding agreement is not a valid strategy when expert data is sparse: imagine a sub-sampled expert trajectory where the agent has the capability to return to previously seen states. In a more typical distribution-matching objective, the agent has to visit all the demonstration states and thus still has to complete the trajectory. In the objective proposed by the authors, the agent can maximize reward by returning to the same demonstration-state repeatedly. Naturally, if demonstrations are available in all states and the agent’s representational capacity is unlimited, then maximizing agreement is trivially optimal. The author’s bound does not require demonstrations to be available in all states and may thus be useful; however, the required amount of expert data is extremely large and no statement can be made if less data is available.

In the related works section, the authors claim that the method is minimizing the GAIL-objective. This is misleading; the GAIL objective is equivalent to minimizing the JS divergence to the expert histogram for all choices of N while the proposed method only minimizes TV distance and thus JS divergence if N is large enough.

For the empirical evaluation, the authors use an L2-distance as a stand-in for the reward used in the analysis. This is necessary as the reward is undefined in continuous state-action spaces (similarly, it will be too sparse in large discrete state-action spaces), but the implications of that heuristic are not explored in the paper. It would be good if the authors could add motivation and discussion for this form of the reward. The results are very positive, but it is unclear whether they are due to the theoretical properties of the algorithm or due to the particular nature of the mujoco walkers. Since the reward is a part of the observation-space in these domains, using simple heuristics like squared distance can plausibly lead to high performance; however, this would not generalize to other domains. It is furthermore problematic that the hardest benchmark task, humanoid, is missing as the given baseline algorithms have no problem solving it. Finally, the reported numbers for behavioral cloning are significantly worse than what has previously been reported in the literature.


**Summary Of The Paper:**

The authors propose an algorithm for imitation learning which rewards the agent for observing state-action pairs that are part of the demonstration-set. The main contribution of the paper is the theoretical analysis which shows that the algorithm, given sufficient expert data, matches the expert’s occupancy distribution and expected reward. The proof comes in three parts: first, the authors relate expert rollouts to the limiting distribution of the expert’s policy. Second, the authors put a probabilistic bound on the expected agreement with expert-data based on the mixing time of the agent. Finally, the authors use this bound on expected agreement to bound the difference in expected extrinsic reward. The authors also provide an empirical evaluation in standard control benchmarks.

**Summary Of The Review:**

The proposed bound is non-trivial and potentially useful; however, it would benefit from additional context to show whether it is truly significant. The empirical results are interesting, but rely on a heuristic that is not well-motivated and may be domain specific.

---

> ### Author Response · Authors · 2021-11-14
> **Thanks for the detailed and thoughtful review.**
>
> Concerning the structure of the paper, we will follow your advice and include lemmas from appendices C & D in the main text.
>
> You say that the amount of data required by our bound grows quadratically in the size of the state space. This is correct. However, we still think that our bound represents a meaningful contribution to the community. Prior to our paper, the algorithm introduced by Wang et al. (2019) has enjoyed no bound at all, being a heuristic. We agree that an algorithm with a sharper result would be nicer, but a bound is better than not having a bound. Moreover, in practice, experiments show that the amount of data needed to imitate experts to an acceptable level is reasonable and often smaller than GAIL and other baseline algorithms. You also say that the approach of rewarding agreement with the expert will have difficulties working with a sub-sampled dataset. Again, this is correct. However, our algorithm is targeting the scenario where expert data is scarce. As such, throwing away steps from a trajectory (as required by sub-sampling) seems wasteful.
>
> Moreover, you say that our claim that the algorithm is minimizing the GAIL objective only holds when N is large enough. Again this is correct and we will clarify this in a revised version of the paper. However, it is also true that the GAIL objective only guarantees recovery of expert reward if the expert state-action histogram is close to the expert state-action distribution (which again requires N to be large, although possibly not as large are required by our bound).
>
> You also mention the problem of how the reward used in the experiments (which is based on the L2-distance) is related to the reward used in the theoretical analysis. To see how, consider a scaled version of the binary reward used in the theoretical analysis which ranges from -L to 1 instead of from 0 to 1 (where L is the L2-diameter of the state-action space). The relaxed reward is an upper bound on the theoretical reward. While we can no longer theoretically claim that optimizing this upper bound guarantees the recovery of expert reward (as in Lemma 4), this property still holds in practice, as verified by the experiment reported in the "Intrinsic and Extrinsic Returns are Related in Practice" paragraph. We agree that this justification should have been included and will do so in the updated version of the paper. Moreover, we are not the only algorithm that requires computation of the L2 distance (GMMIL has this requirement as well). Ultimately, we do not claim that our reward relaxation works for every domain or even that it is the best way to relax the original theoretical rewards. In other settings, the random expert distillation approach can be used, which Wang et al. (2019) have demonstrated can work quite well and Burda et al. (2018) have shown can work with pixel observations.
>
> Moreover, you say it is problematic that the humanoid domain is not included. This is for two reasons. First, we wanted to use an established benchmark suite. We went with d4rl-PyBullet (https://github.com/takuseno/d4rl-pybullet), which does not include humanoid. Second, the PyBullet version of humanoid (as opposed to MuJoCo) is known to have bugs (https://pybullet.org/Bullet/phpBB3/viewtopic.php?t=13232, https://www.youtube.com/watch?v=KrU34hSWH_E), which make it hard to use as a reliable benchmark.
>
> Finally, you say that our numbers for BC are worse than reported in literature. This is because of the difference between the PyBullet and the MuJoCo environments. Despite efforts by the authors of PyBullet-gym, scores obtained with MuJoCo and PyBullet are not comparable. We chose PyBullet because it was the only open source solution available at the time we were running the experiments.
>
> [1] Exploration by Random Network Distillation Yuri Burda, Harrison Edwards, Amos Storkey, Oleg Klimov

---

> > ### Comment · Reviewer_TFjU · 2021-11-22
> > **Re:**
> >
> > Thanks for your response,
> >
> > regarding the bounds, I agree that it is useful to have it.
> >
> > regarding sub-sampling trajectories, it is indeed just an experimental crutch that points at control domains being to easy for imitation learning - the real phenomenon is that an imperfect agent will stray from the demonstrations which may lead to the same phenomenon.
> >
> > regarding experiments, I agree that the pybullet experiments without Humanoid should be acceptable given the date of submission. I am still worried that the reward relaxation is too specific to the given environments. It is usually possible to train good control policies in observation spaces that exclude the extrinsic reward, i.e. x-velocity. This could be a useful experiment to convince people that the approach is general. Methods that are truly imitating the expert, like GAIL, will work on this out of the box, but some published imitation learning methods can be shown to be over-regularized in this way.

---

> > > ### Author Response · Authors · 2021-11-22
> > > **Thanks for the comment.**
> > >
> > > The new ablation you suggested (repeating the experiment with a partially obscured observation space where the velocity along the x axis is missing) seems to target POMDPs. We wanted to stress that our method is designed for MDPs, not POMDPs. We believe that the existing experiments are enough to verify it works for MDPs.

---

### Author Response · Authors · 2021-11-21
**Paper Revision**

We have posted a new revision of the paper, incorporating feedback from the reviewers. Changes include:
- we added a justification of the continuous relaxation of the reward
- we integrated lemmas about Total Variation and Markov chain mixing into the main text
- we clarified GAIL equivalence only holds for large N
- we added a run-time comparison of different algorithms

as well as many smaller clarifications.

Our changes mean that the numbering of the lemmas has changed.

---

### Public Comment · ~Kamil_Ciosek1 · 2022-05-09
**Source Code**

The source code for the paper is available at https://github.com/spotify-research/il-by-rl.

---

### Decision · Program_Chairs · 2022-01-20

**Decision:**

Accept (Poster)

**Comment:**

The paper provides theoretical bounds for imitation learning with rewards (algorithm from Wang et al. (2019)). The bounds/proofs are highly novel and a very interesting contribution to the community, even though they are a lot more conservative than what is observed in practice. All reviewers agree on this point.
It is laudable that the authors also additionally provide an experimental evaluation. After the revision and the discussion, quite a few of the reviewers are still not 100% convinced about them, on the one hand as they would have liked to see more tasks, and on the other hand due to concerns about the reward relaxation (i.e., doesn't match the assumptions in the theorems any longer) which is required for experiments on standard benchmarks.
In the final answer the authors provide evidence that there is no big discrepancy, which is good enough (given that there don't seem to be any alternatives to get around this issue, except removing the experimental section altogether, which would be undesirable). Please clearly point out those limitations of the experiments in the paper and also incorporate this evidence.